# Sensorless LC Filter Implementation for Permanent Magnet Machine Drive Using Observer-Based Voltage and Current Estimation

**DOI:** 10.3390/s21113596

**Published:** 2021-05-21

**Authors:** Chia-Ming Liang, Yi-Jen Lin, Jyun-You Chen, Guan-Ren Chen, Shih-Chin Yang

**Affiliations:** Department of Mechanical Engineering, National Taiwan University, No.1, Sec. 4, Roosevelt Road, Taipei 10617, Taiwan; r08522825@ntu.edu.tw (C.-M.L.); f08522816@ntu.edu.tw (Y.-J.L.); d06522004@ntu.edu.tw (J.-Y.C.); d03522014@ntu.edu.tw (G.-R.C.)

**Keywords:** low inductance PM machine, PWM inverter filter, observer-based estimation

## Abstract

For pulse width modulation (PWM) inverter drives, an LC filter can cascade to a permanent magnet (PM) machine at inverter output to reduce PWM-reflected current harmonics. Because the LC filter causes resonance, the filter output current and voltage are required for the sensorless field-oriented control (FOC) drive. However, existing sensors and inverters are typically integrated inside commercial closed-form drives; it is not possible for these drives to obtain additional filter output signals. To resolve this integration issue, this paper proposes a sensorless LC filter state estimation using only the drive inside current sensors. The design principle of the LC filter is first introduced to remove PWM current harmonics. A dual-observer is then proposed to estimate the filter output current and voltage for the sensorless FOC drive. Compared to conventional model-based estimation, the proposed dual-observer demonstrates robust estimation performance under parameter error. The capacitor parameter error shows a negligible influence on the proposed observer estimation. The filter inductance error only affects the capacitor current estimation at high speed. The performance of the sensorless FOC drive using the proposed dual-observer is comparable to the same drive using external sensors for filter voltage and current measurement. All experiments are verified by a PM machine with only 130 μH phase inductance.

## 1. Introduction

Compared to conventional industrial PM machines, low-inductance machines have a lower inductive voltage and lower back electromotive force (EMF) voltage drop. These machines are suitable for high-speed applications under volume limitation; e.g., drones, spindles and compressors [1,2,3]. However, considering PWM voltage control, visible switching-reflected current harmonics occur on these low-inductance machines [4]. Although the bus voltage usage is improved at high speed, PWM inverter-reflected current harmonics increase the machine iron loss and cause rotor thermal issues [5,6].

LC filters can be installed in PM machine drives for the reduction of PWM current harmonics. Because LC filters do not include resistances, there is a resulting voltage and current distortion, especially for operation near filter resonance. Circulating currents are induced between the inverter and LC filter. In this case, active damping compensation is used to minimize the filter-reflected current/voltage distortion under the FOC drive. In [7,8,9,10], a cascaded controller with both an external current loop and an internal voltage loop is proposed in an induction machine for active damping compensation. However, three voltage sensors and two current sensors are required, with considerable cost. The authors of [11,12,13,14] simplify this cascaded control topology by using a simple current controller. In contrast to the voltage and current control in [7,8,9], only either the capacitor voltage or the current is closed-loop regulated. In this case, the LC filter parameters are required to maintain the active damping performance. Furthermore, for high-speed machine operation, the LC filter causes additional phase delay on machine currents. In [15,16], a discretized voltage model is proposed for digital delay compensation. A PWM inverter can achieve the current regulation on an RL load with 400 Hz frequency. Among these active damping compensations, it is noted that capacitor currents/voltages and inductor output machine currents are required for the FOC drive and active damping compensation.

For a sensorless machine drive, the inductor output machine current and capacitor voltage must be known for machine current regulation and rotor position estimation. The filter output voltage and current can be obtained by external sensors or estimated by state observers. In general, current/voltage sensors and inverters are typically integrated inside commercial closed-form drives. Additional inverter hardware modification is necessary if external sensors are selected for an FOC drive with an LC filter [17,18]. In contrast, sensorless observer-based estimation is preferred because only the inverter inside the current sensors is used for filter state estimation. In [19], the capacitor current is estimated using an LC filter observer based on the filter output current measurement. Although no voltage sensor is present, three current sensors outside the inverter are added for the observer estimation. To position sensorless FOC drives, the LC filter is also attached for current regulation [5]. However, similar to [19], filter output currents are measured in order to estimate the machine EMF voltage. In [20], with an LC filter, the direct torque and flux control is implemented for the machine drive. Considering the filter effect on the drive, several corrections are added to correct flux and torque commands. More importantly, this direct torque control still requires filter output currents to estimate the machine flux. From the review of these LC filter-related works, the integration of a filter on closed-form drives without external current sensors is desired for high-performance machine control.

This paper proposes a sensorless LC filter implementation for an FOC drive using only the inverter inside current sensors for the filter current and voltage estimation. In [21], a sensorless LC filter state estimation using only the closed-form drive inside current sensors is developed; however, only the preliminary observer topology is illustrated, neglecting various LC filter cross-coupling effects. These coupling issues including the parameter sensitivity on the filter state estimation are fully investigated here. In this paper, a dual-observer topology is proposed to estimate the filter inductor current and capacitor voltage for both current regulation and position estimation, respectively. Compared to conventional model-based filter voltage and current estimation, the proposed dual-observer can reduce the filter parameter sensitivity, resulting in better state estimation performance. Based on experimental results, a comparable FOC drive is developed and compared to the same drive using external sensors for filter voltage and current measurement. All the tests are verified on a 105 W PM machine with only 130 μH phase inductance.

## 2. LC Filter Design Principle

This section explains the selection of the LC filter on the PM machine FOC drive. Figure 1 illustrates the overall PWM inverter including an output LC filter. For the PWM machine drive, the LC filter along with the machine stator windings can be viewed as an LCL filter. To remove PWM-reflected current harmonics on the machine, the equivalent resonant frequency f_filter_ of the LCL filter can be selected based on the critical resonance frequency f_crit_. As reported in [22], f_crit_ is suggested by
(1)$$f_{\mathrm{crit}}=\frac{f_{\mathrm{sample}}}{6}=\frac{f_{\mathrm{PWM}}}{6}\;(\mathrm{assuming}\;f_{\mathrm{sample}}=\;f_{\mathrm{PWM}})$$
where f_sample_ is the controller sample frequency synchronized to PWM frequency f_PWM_. To maintain the dynamic response of machine control, the resonant frequency f_filter_ should be designed higher than f_crit_. On the other hand, f_filter_ must be lower than half of f_sample_ for the purpose of reducing the PWM-reflected current harmonics. In this paper, f_sample_ and f_crit_ are 10 kHz and 1.667 kHz, respectively. However, the high f_crit_ might cause a challenge regarding the filter state estimation for the proposed senseless integration of the LC filter and FOC drive. Considering the available specification of inductances and capacitors, the resonant frequency f_filter_ is selected at 2.9 kHz based on the above criteria.

After the determination of f_filter_, the individual filter inductance and capacitor value are then selected according to the requirements of the machine phase current ripples. In [23,24], the relationship between the inductor current ripple ΔI_L_ and filter inductance can be shown to be
(2)$$\Delta I_{L}=\frac{1}{8}\frac{V_{C}}{Lf_{sample}}$$
where Vc is the corresponding capacitor voltage. To reach a tolerable current ripple of around 5%, the filter inductance is designed at 1.2 mH, where the resulting capacitor is 25.8 μF. Considering the accessibility of LC components in the market, the filter inductance is re-selected at 1 mH, giving a resonant frequency of 2.92 kHz and current ripple of around 6.31%.

## 3. Machine Drive with LC Filter

This section explains the improvement of the low-inductance PM machine using the LC filter to reduce PWM current harmonics. For variable-frequency drive applications, the positioning of the senseless FOC drive is applied. As seen for the proposed machine drive with an LC filter in Figure 1, I_LA_/I_LB_/I_LC_ are the inverter A/B/C-phase currents across three filter inductances, respectively, I_CA_/I_CB_/I_CC_ are the phase currents across filter capacitors, and V_CA_/V_CB_/V_CC_ are the phase voltages on filter capacitors. Besides, I_A_/I_B_/I_C_ are the inductor output currents flowing across the machine phase windings.

It is noteworthy that, for standard machine drives, only the currents of I_LA_/I_LB_/I_LC_ output from the inverter can be measured using existing current sensors. Considering the influence of the LC filter, machine phase voltages and currents can be affected by the capacitor voltages V_CA_/V_CB_/V_CC_ and currents I_CA_/I_CB_/I_CC_. In this case, the PM machine model can be modified by
(3)VCA=Rs(ILA−ICA)+Lsd(ILA−ICA)dt+ωeλpm_AVCB=Rs(ILB−ICB)+Lsd(ILB−ICB)dt+ωeλpm_BVCC=Rs(ILC−ICC)+Lsd(ILC−ICC)dt+ωeλpm_C
where R_s_ and L_s_ are the PM machine phase resistance and inductance, respectively, assuming no saliency for simplicity, and λ_pm_A_/λ_pm_B_/λ_pm_C_ are the corresponding phase magnet flux linkages. For the senseless FOC drive, the knowledge of capacitor currents I_CA_/I_CB_/I_CC_ and voltages V_CA_/V_CB_/V_CC_ is required for both the current regulation and EMF-based position estimation. Table 1 lists the test PM machine specification. The rated speed is 42 krpm with 700 Hz electric frequency. It is noted that the phase inductance is only 130 μH. The visible PWM current harmonics are expected during the operation of the FOC drive. In the next section, the observer-based estimation of capacitor voltages and currents is explained using only the available measurable current signals for the drive: I_LA_/I_LB_/I_LC_.

## 4. Filter Voltage and Current Estimation

This section explains the estimation of the capacitor currents I_CA_/I_CB_/I_CC_ and voltages V_CA_/V_CB_/V_CC_ using the proposed dual-observer. Figure 2 demonstrates the signal process for the proposed senseless FOC drive with an LC filter. Considering the operation of the FOC drive at high speed, the proposed dual-observer is developed in a dq machine rotor-referred synchronous frame. In this case, all voltages/currents are transferred to DC signals, which are suited for closed-loop current regulation. This dual-observer consists of both a capacitor voltage observer and capacitor current observer for the estimation of ${\hat{V}}_{\mathrm{Cdq}}$, ${\hat{I}}_{\mathrm{Cdq}}$ and ${\hat{I}}_{\mathrm{dq}}$. For the senseless FOC drive, the estimated machine dq currents ${\hat{I}}_{\mathrm{dq}}$ are used for the current regulation. Besides, the estimated capacitor voltages ${\hat{V}}_{\mathrm{Cdq}}$ should be applied for machine EMF estimation and thus rotor position estimation.

### 4.1. Capacitor Voltage Estimation

Figure 2 (bottom-left) denotes the proposed capacitor voltage observer. Assuming that the inverter deadtime is negligible, the inverter voltage commands ${\overset{*}{V}}_{\mathrm{dq}}$ can contribute to both the capacitor voltages V_Cdq_ and inductor voltages V_Ldq_. Under this effect, the dynamic model of capacitor voltage is illustrated by
(4)[VCdVCq]=[Vd∗Vq∗]−[VLdVLq]=[Vd∗Vq∗]−[RfILd+LfdILddt−LfωeILqRfILq+LfdILqdt+LfωeILd]
Based on (4), the capacitor voltage V_Cdq_ can be directly estimated by
(5)[V^CdV^Cq]=[Vd∗Vq∗]−[R^fILd+L^fdILddt−L^fω^eILqR^fILq+L^fdILqdt+L^fω^eILd]
where ${\hat{V}}_{\mathrm{Ldq}}$ is the estimated dq inductance voltage, $\hat{L_{f}}$ and $\hat{R_{f}}$ are the estimated filter inductance and parasitic resistance, and $\hat{\omega_{e}}$ is the estimated rotor speed. Equation (6) denotes the estimation accuracy of ${\hat{V}}_{\mathrm{Cdq}}$ in the S-domain for the following parameter sensitivity analysis comparing to the proposed observer estimation.
(6)$$\left[\begin{array}{c} {\hat{V}}_{Cd\_err}\left(s\right)\\ {\hat{V}}_{Cq\_err}\left(s\right) \end{array}\right]=\left[\begin{array}{c} V_{Cd}\left(s\right)-{\hat{V}}_{Cd}\left(s\right)\\ V_{Cq}\left(s\right)-{\hat{V}}_{Cq}\left(s\right) \end{array}\right]\;=-\left[\begin{array}{c} \left(R_{f}-{\hat{R}}_{f}\right)-\left(L_{f}-{\hat{L}}_{f}\right)s\\ \left(R_{f}-{\hat{R}}_{f}\right)-\left(L_{f}-{\hat{L}}_{f}\right)s \end{array}\right]\left[\begin{array}{c} I_{Ld}\left(s\right)\\ I_{Lq}\left(s\right) \end{array}\right]+\left[\begin{array}{c} -\left(L_{f}\omega_{e}-{\hat{L}}_{f}{\hat{\omega }}_{e}\right)\\ \left(L_{f}\omega_{e}-{\hat{L}}_{f}{\hat{\omega }}_{e}\right) \end{array}\right]\left[\begin{array}{c} I_{Lq}\left(s\right)\\ I_{Ld}\left(s\right) \end{array}\right]$$
It is noted that the error of $V_{\mathrm{Cdq\_err}}\;=\;V_{\mathrm{Cdq}}\;-\;{\hat{V}}_{\mathrm{Cdq}}$ in Equation (6) is designed to analyze the accuracy of ${\hat{V}}_{\mathrm{Cdq}}$. Considering perfect parameters of $\hat{R_{f}}\;=\;R_{f}$ and $\hat{L_{f}}\;=\;L_{f}$, V_Cdq_err_ should be zero for all operation conditions. Figure 3 simulates the estimation accuracy of |V_Cdq_err_(jω)/I_Ldp_(jω)|. In this figure, different absolute magnitudes under different operating frequencies are calculated. The parameter errors of $\hat{R_{f}}$ and $\hat{L_{f}}$ are calculated based on Equation (6). As seen in Figure 3a, V_Cdq_err_ is strongly affected by the $\hat{R_{f}}$ error independent of the operating frequencies. In contrast, in (b), the influence of the $\hat{L_{f}}$ error on V_Cdq_err_ increases as frequency increases. Besides, in (c), the inductor cross-coupling voltage also results in a constant error of V_Cdq_err_ independent of the operating frequency. Considering the direct capacitor voltage estimation in Equation (5), it is concluded that ${\hat{V}}_{\mathrm{Cdq}}$ estimation accuracy is strongly dependent on the inductor parameters $\hat{R_{f}}$ and $\hat{L_{f}}$.

In this paper, the closed-loop observer is developed to reduce the parameter sensitivity through the feedback control of I_Ldq_. Figure 4 illustrates the first closed-loop observer in the S-domain for the capacitor voltage estimation. In this observer, the inputs are the inductor-measured dq currents I_Ldq_ while the observer outputs are the estimated dq capacitor voltages ${\hat{V}}_{\mathrm{Cdq}}$. Besides, the observer controller C_V_(s) and feedforward voltage inputs $V_{\mathrm{CFF\_V}}^{*}$ are designed based on
(7)$$C_{V}(s)=\frac{K_{p\_V}s+K_{i\_V}}{s}$$
(8)$$\left[\begin{array}{c} V_{CFF\_Vd}^{*}\\ V_{CFF\_Vq}^{*} \end{array}\right]=\left[\begin{array}{c} V_{d}^{*}\\ V_{q}^{*} \end{array}\right]+\left[\begin{array}{c} {\hat{V}}_{cross\_d}\\ {\hat{V}}_{cross\_q} \end{array}\right]=\left[\begin{array}{c} V_{d}^{*}\\ V_{q}^{*} \end{array}\right]+\left[\begin{array}{c} {\hat{L}}_{f}{\hat{\omega }}_{e}I_{Lq}\\ -{\hat{L}}_{f}{\hat{\omega }}_{e}I_{Ld} \end{array}\right]$$
where K_p_V_ and K_i_V_ are the proportional and integral gain of the closed-loop observer, respectively. Based on closed-loop regulation, ${\hat{V}}_{\mathrm{Cdq}}$ can be modeled by the external voltage disturbance. Under this effect, the capacitor voltages V_Cdq_ can be estimated by
(9)$$\left[\begin{array}{c} {\hat{V}}_{Cd}(s)\\ {\hat{V}}_{Cq}(s) \end{array}\right]=K_{p\_V}\left[\begin{array}{c} I_{Ld}(s)-{\hat{I}}_{Ld}(s)\\ I_{Lq}(s)-{\hat{I}}_{Lq}(s) \end{array}\right]+\frac{K_{i\_V}}{s}\left[\begin{array}{c} I_{Ld}(s)-{\hat{I}}_{Ld}(s)\\ I_{Ld}(s)-{\hat{I}}_{Ld}(s) \end{array}\right]$$
As seen from the signal process in Figure 4, the overall dynamic model of ${\hat{V}}_{\mathrm{Cdq}}$ can be formulated by
(10)$$\left[\begin{array}{c} {\hat{V}}_{Cd}(s)\\ {\hat{V}}_{Cq}(s) \end{array}\right]=\left(\frac{K_{p\_V}s+K_{i\_V}}{s}\right)\left(\frac{L_{f}s^{2}+\left(R_{f}+K_{p\_V}\right)s+K_{i\_V}}{{\hat{L}}_{f}s^{2}+\left({\hat{R}}_{f}+K_{p\_V}\right)s+K_{i\_V}}-1\right)\left[\begin{array}{c} I_{Ld}(s)\\ I_{Lq}(s) \end{array}\right]\;+\frac{K_{p\_V}s+K_{i\_V}}{{\hat{L}}_{f}s^{2}+\left({\hat{R}}_{f}+K_{p\_V}\right)s+K_{i\_V}}\left({\hat{L}}_{f}{\hat{\omega }}_{e}-L_{f}\omega_{e}\right)\left[\begin{array}{c} I_{Lq}(s)\\ -I_{Ld}(s) \end{array}\right]\;+\frac{K_{p\_V}s+K_{i\_V}}{{\hat{L}}_{f}s^{2}+\left({\hat{R}}_{f}+j\omega_{e}{\hat{L}}_{f}+K_{p\_V}\right)s+K_{i\_V}}\left[\begin{array}{c} V_{Cd}(s)\\ V_{Cq}(s) \end{array}\right]$$

In Equation (10), the estimated ${\hat{V}}_{\mathrm{Cdq}}$ consists of three terms: the first two terms are proportional to I_Ldq_. If both $\hat{R_{f}}$ = R_f_ and $\hat{L_{f}}$ = L_f_, these two terms disappear. In contrast, the third term is proportional to the actual capacitor voltages V_Cdq_. This term is independent of the parameter errors at low frequency. Considering the filter parameter errors, the estimation accuracy of proposed observer can be represented by
(11)$$\left[\begin{array}{c} V_{Cd\_err}(s)\\ V_{Cq\_err}(s) \end{array}\right]=\left(\frac{K_{p\_V}s+K_{i\_V}}{s}\right)\left(\frac{L_{f}s^{2}+\left(R_{f}+K_{p\_V}\right)s+K_{i\_V}}{{\hat{L}}_{f}s^{2}+\left({\hat{R}}_{f}+K_{p\_V}\right)s+K_{i\_V}}-1\right)\left[\begin{array}{c} I_{Ld}(s)\\ I_{Lq}(s) \end{array}\right]\;+\frac{K_{p\_V}s+K_{i\_V}}{{\hat{L}}_{f}s^{2}+\left({\hat{R}}_{f}+K_{p\_V}\right)s+K_{i\_V}}\left({\hat{L}}_{f}{\hat{\omega }}_{e}-L_{f}\omega_{e}\right)\left[\begin{array}{c} I_{Lq}(s)\\ -I_{Ld}(s) \end{array}\right]$$

Comparing the proposed estimation error in Equation (11) to open-loop direct estimation in Equation (6), the estimation accuracy of ${\hat{V}}_{\mathrm{Cdq}}$ is related to both the parameters $\hat{R_{f}}/\hat{L_{f}}$ and controller gains K_p_V_/K_i_V_. By properly designing K_p_V_ and K_i_V_, it is clear that the parameter sensitivity on ${\hat{V}}_{\mathrm{Cdq}}$ can be improved.

Figure 3 also illustrates the same estimation accuracy of |V_Cdq_err_(jω)/I_Ldp_(jω)| based on the proposed closed-loop voltage observer. In this simulation, K_p_V_ and K_i_V_ are selected to achieve a 100 Hz bandwidth considering high-frequency noise reduction. Considering the $\hat{R_{f}}$ parameter error in Figure 3a, V_Cdq_err_ is reduced when the operating frequency is beyond the observer bandwidth, as presented by the dashed lines. Besides, for the $\hat{L_{f}}$ error resulting from self-inductance in (b) and cross-inductance in (c), V_Cd_err_ also maintains constant values or decreases beyond the observer bandwidth. By applying the proposed voltage observer in Figure 4, it is shown that the influence of filter parameter errors on ${\hat{V}}_{\mathrm{Cdq}}$ is reduced due to the closed-loop control regulation.

### 4.2. Capacitor Current Estimation

This section explains the capacitor current I_Cdq_ and machine current I_dq_ estimation using the proposed capacitor current observer in Figure 2. On the basis, the capacitor current I_Cdq_ can be estimated through the capacitor model. It is shown by
(12)[ICdICq]=[ILdILq]−[IdIq]=[CpdVCddt−CpωeVCqCpdVCqdt+CpωeVCd]
In Equation (12), the machine currents can also be obtained once capacitor currents are estimated, whereby ${\hat{I}}_{\mathrm{dq}}\;=\;I_{\mathrm{Ldq}}\;-\;{\hat{I}}_{\mathrm{Cdq}}$. Thus, the proposed current observer is focused on ${\hat{I}}_{\mathrm{Cdq}}$ for simplicity. A straightforward way of achieving ${\hat{I}}_{\mathrm{Cdq}}$ estimation can be developed based on Equation (13).
(13)[I^CdI^Cq]=[C^pdV^Cddt−C^pω^eV^CqC^pdV^Cqdt+C^pω^eV^Cd]
where $\hat{C_{p}}$ is the estimated capacitor value. In Equation (13), ${\hat{I}}_{\mathrm{Cdq}}$ estimation is dependent on both $\hat{C_{p}}$ and ${\hat{V}}_{\mathrm{Cdq}}$, which are estimated from the prior voltage observer. As mentioned in Figure 3, the accuracy of ${\hat{V}}_{\mathrm{Cdq}}$ is dependent on the inductor parameters ${\hat{R}}_{f}$ and ${\hat{L}}_{f}$. In order to evaluate the influence of ${\hat{R}}_{f}$ and ${\hat{L}}_{f}$ errors on ${\hat{I}}_{\mathrm{Cdq}}$, ${\hat{V}}_{\mathrm{Cdq}}$ in Equation (13) is replaced by Equation (5), as derived by
(14)[I^CdI^Cq]=C^p[dVd∗dt−ω^eVq∗dVq∗dt+ω^eVd∗]+C^pR^f[−dILddt−dILqdt]+C^pL^f[−d2ILddt2+ω^edILqdt−d2ILqdt2−ω^edILddt]+C^pR^f[ω^eILq−ω^eILd]+C^pL^f[ω^edILqdt+ω^e2ILd−ω^edILddt−ω^e2ILq]

Similar to Equation (6), the ${\hat{I}}_{\mathrm{Cdq}}$ estimation accuracy can be analyzed in the S-domain based on the definition of the capacitor current error, ${\hat{I}}_{\mathrm{Cdq\_err}}\;=\;I_{\mathrm{Cdq}}\;-\;{\hat{I}}_{\mathrm{Cdq}}$. It is shown to be
(15)$$\begin{array}{cc} \left[\begin{array}{c} I_{Cd\_err}(s)\\ I_{Cq\_err}(s) \end{array}\right]= & \left[\begin{array}{c} I_{Cd}(s)-{\hat{I}}_{Cd}(s)\\ I_{Cq}(s)-{\hat{I}}_{Cq}(s) \end{array}\right]=\left(C_{p}-{\hat{C}}_{p}\right)\left[\begin{array}{c} sV_{d}^{*}-\omega_{e}V_{q}^{*}\\ sV_{q}^{*}+\omega_{e}V_{d}^{*} \end{array}\right]\\ & +\left(C_{p}R_{f}-{\hat{C}}_{p}{\hat{R}}_{f}\right)\left[\begin{array}{c} -sI_{Ld}\\ -sI_{Lq} \end{array}\right]+\left(C_{p}L_{f}-{\hat{C}}_{p}{\hat{L}}_{f}\right)\left[\begin{array}{c} -s^{2}I_{Ld}+s\omega_{e}I_{Lq}\\ -s^{2}I_{Lq}-s\omega_{e}I_{Ld} \end{array}\right]\\ & +\left(C_{p}R_{f}-{\hat{C}}_{p}{\hat{R}}_{f}\right)\left[\begin{array}{c} \omega_{e}I_{Lq}\\ -\omega_{e}I_{Ld} \end{array}\right]+\left(C_{p}L_{f}-{\hat{C}}_{p}{\hat{L}}_{f}\right)\left[\begin{array}{c} s\omega_{e}I_{Lq}+\omega_{e}^{2}I_{Ld}\\ -s\omega_{e}I_{Ld}-\omega_{e}^{2}I_{Lq} \end{array}\right] \end{array}$$
Here, there is no speed estimation error, where $\hat{\omega_{e}}\;=\;\omega_{e}$ is assumed for simplicity. It is found that I_Cdq_err_ is proportional to both the measured inductor current I_Ldq_ and inverter voltage command $V_{\mathrm{dq}}^{*}$. In this case, the estimation accuracy of I_Cdq_err_ with respect to either I_Ldq_ and $V_{\mathrm{dq}}^{*}$ is investigated. Figure 5 demonstrates the estimation accuracy of |I_Cdq_err_(jω)/I_Ldp_(jω)| and |I_Cdq_err_(jω)/$V_{\mathrm{dq}}^{*}$(jω)| based on Equation (15) considering parameter errors on $\hat{L_{f}}$ and $\hat{C_{p}}$. In this sensitivity analysis, the $\hat{R_{f}}$ error is excluded for simplicity. As seen in Equation (15), the $\hat{R_{f}}$ error also affects ${\hat{I}}_{\mathrm{Cdq}}$ estimation dependent on the magnitude of I_Ldq_. However, considering the laminated core inductance, the parasitic resistance is sufficiently low and can be neglected.

Figure 5 illustrates the $\hat{L_{f}}$ error regarding the d-axis capacitor current estimation error I_Cdq_err_ resulting from the self-coupled d-axis inductor current I_Ld_ and cross-coupled q-axis current I_Lq_, respectively. As seen from |I_Cd_err_(jω)/I_Ld_(jω)| in (a), a constant error occurs at low frequency. In contrast, for |I_Cd_err_(jω)/I_Lq_(jω)| in (b), the estimation error increases as frequency increases. Besides, Figure 5c shows the $\hat{C_{p}}$ error on I_Cd_err_ caused by the self-coupled d-axis voltage command $V_{d}^{*}$. As seen from Equation (15), the estimation error also increases as frequency increases. A similar result can be found for the $\hat{L_{f}}$ error on cross-coupled |I_Cd_err_(jω)/I_Lq_(jω)|. Based on the sensitivity analysis on the direct ${\hat{I}}_{\mathrm{Cdq}}$ estimation in Equation (13), the $\hat{L_{f}}$ error results in a constant estimation error on ${\hat{I}}_{\mathrm{Cdq}}$ at low frequency, and more importantly, the estimation error increases as the operating frequency increases. Below, a closed-loop observer-based current estimation is proposed to reduce this parameter sensitivity.

Similar to the capacitor voltage observer, the capacitor current I_Cdq_ can be estimated through another closed-loop observer to improve the estimation performance. Figure 6 proposes another current observer for both I_Cdq_ and machine current I_dq_ estimation. In this observer, the input is the estimated capacitor voltage ${\hat{V}}_{\mathrm{Cdq}}$ from the first observer in Figure 4, and the outputs are the estimated capacitor current ${\hat{I}}_{\mathrm{Cdq}}$ and machine current ${\hat{I}}_{\mathrm{dq}}$. It is noted that ${\hat{I}}_{\mathrm{dq}}$ is useful for the machine current regulation of the FOC drive, while ${\hat{I}}_{\mathrm{Cdq}}$ can be applied for the active damping compensation of the LC filter.

For the proposed current observer in Figure 6, the dynamic model is related to the estimated voltage ${\hat{V}}_{\mathrm{Cdq}}$ and voltage feedforward $V_{\mathrm{CFF\_Idq}}^{*}$. First, the capacitor voltage model in dq frame can be derived in the S-domain by
(16)$$\left[\begin{array}{c} {\hat{I}}_{Cd}\\ {\hat{I}}_{Cq} \end{array}\right]=\left[\begin{array}{c} I_{Ld}-{\hat{I}}_{d}\\ I_{Lq}-{\hat{I}}_{q} \end{array}\right]=\left[\begin{array}{c} {\hat{C}}_{p}s{\hat{V}}_{Cd}-{\hat{C}}_{p}{\hat{\omega }}_{e}{\hat{V}}_{Cq}\\ {\hat{C}}_{p}s{\hat{V}}_{Cq}+{\hat{C}}_{p}{\hat{\omega }}_{e}{\hat{V}}_{Cd} \end{array}\right]$$
After that, the inductor currents I_Ldq_ in Equation (16) can be replaced from both $V_{\mathrm{dq}}^{*}$ and ${\hat{V}}_{\mathrm{Cdq}}$ in Equation (5). In this case, the resulting dynamic model is shown by
(17)$$\left[\begin{array}{c} \left[{\hat{L}}_{f}{\hat{C}}_{p}s^{2}+{\hat{R}}_{f}{\hat{C}}_{p}s+\left(1-{\hat{\omega }}_{e}{\hat{L}}_{f}{\hat{C}}_{p}\right)\right]{\hat{V}}_{Cd}\\ \left[{\hat{L}}_{f}{\hat{C}}_{p}s^{2}+{\hat{R}}_{f}{\hat{C}}_{p}s+\left(1-{\hat{\omega }}_{e}{\hat{L}}_{f}{\hat{C}}_{p}\right)\right]{\hat{V}}_{Cq} \end{array}\right]=\left[\begin{array}{c} V_{d}^{*}+\left(2{\hat{L}}_{f}{\hat{C}}_{p}{\hat{\omega }}_{e}s+{\hat{R}}_{f}{\hat{C}}_{p}{\hat{\omega }}_{e}\right){\hat{V}}_{Cq}-\left({\hat{R}}_{f}+{\hat{L}}_{f}s\right){\hat{I}}_{d}+{\hat{L}}_{f}{\hat{\omega }}_{e}{\hat{I}}_{q}\\ V_{q}^{*}-\left(2{\hat{L}}_{f}{\hat{C}}_{p}{\hat{\omega }}_{e}s+{\hat{R}}_{f}{\hat{C}}_{p}{\hat{\omega }}_{e}\right){\hat{V}}_{Cd}-\left({\hat{R}}_{f}+{\hat{L}}_{f}s\right){\hat{I}}_{q}-{\hat{L}}_{f}{\hat{\omega }}_{e}{\hat{I}}_{d} \end{array}\right]$$
It should be noted in Equation (17) that ${\hat{I}}_{\mathrm{dq}}$ is the desired estimator. In order to formulate the current observer in Figure 6 with voltage regulation, Equation (17) is organized as
(18)$$\left[\begin{array}{l} {\hat{V}}_{dis\_d}\\ {\hat{V}}_{dis\_q} \end{array}\right]+[\begin{array}{c} V_{CFF\_Id}^{*}\\ V_{CFF\_Iq}^{*} \end{array}]=\left[\begin{array}{l} {\hat{L}}_{f}{\hat{C}}_{p}s^{2}+{\hat{R}}_{f}{\hat{C}}_{p}s+(1-{\hat{\omega }}_{e}^{2}{\hat{L}}_{f}{\hat{C}}_{p})\\ {\hat{L}}_{f}{\hat{C}}_{p}s^{2}+{\hat{R}}_{f}{\hat{C}}_{p}s+(1-{\hat{\omega }}_{e}^{2}{\hat{L}}_{f}{\hat{C}}_{p}) \end{array}\right]\left[\begin{array}{c} {\hat{V}}_{Cd}\\ {\hat{V}}_{Cq} \end{array}\right]$$
where the observer internal disturbances ${\hat{V}}_{\mathrm{dis}\_\mathrm{dq}}$ and total feedforward $V_{\mathrm{CFF\_Idq}}^{*}$ are equivalent to
(19)$$\left[\begin{array}{l} {\hat{V}}_{dis\_d}\\ {\hat{V}}_{dis\_q} \end{array}\right]=\left[\begin{array}{l} -{\hat{R}}_{f}{\hat{I}}_{d}-{\hat{L}}_{f}s{\hat{I}}_{d}+{\hat{L}}_{f}{\hat{\omega }}_{e}{\hat{I}}_{q}\\ -{\hat{R}}_{f}{\hat{I}}_{q}-{\hat{L}}_{f}s{\hat{I}}_{q}-{\hat{L}}_{f}{\hat{\omega }}_{e}{\hat{I}}_{d} \end{array}\right]\approx \left[\begin{array}{l} -{\hat{R}}_{f}{\hat{I}}_{d}+{\hat{L}}_{f}{\hat{\omega }}_{e}{\hat{I}}_{q}\\ -{\hat{R}}_{f}{\hat{I}}_{q}-{\hat{L}}_{f}{\hat{\omega }}_{e}{\hat{I}}_{d} \end{array}\right]$$
(20)$$\left[\begin{array}{c} V_{CFF\_Id}^{*}\\ V_{CFF\_Iq}^{*} \end{array}\right]=[\begin{array}{c} V_{d}^{*}+\left(2{\hat{L}}_{f}{\hat{C}}_{p}{\hat{\omega }}_{e}s+{\hat{R}}_{f}{\hat{C}}_{p}{\hat{\omega }}_{e}\right){\hat{V}}_{Cq}\\ V_{q}^{*}-\left(2{\hat{L}}_{f}{\hat{C}}_{p}{\hat{\omega }}_{e}s+{\hat{R}}_{f}{\hat{C}}_{p}{\hat{\omega }}_{e}\right){\hat{V}}_{Cd} \end{array}]\approx [\begin{array}{c} V_{d}^{*}+{\hat{R}}_{f}{\hat{C}}_{p}{\hat{\omega }}_{e}{\hat{V}}_{Cq}\\ V_{q}^{*}-{\hat{R}}_{f}{\hat{C}}_{p}{\hat{\omega }}_{e}{\hat{V}}_{Cd} \end{array}]=[\begin{array}{c} V_{d}^{*}\\ V_{q}^{*} \end{array}]+[\begin{array}{c} {\hat{V}}_{Cd\_cross}\\ {\hat{V}}_{Cq\_cross} \end{array}]$$
In Equation (19) and Equation (20), two differential terms are neglected for simplicity, assuming nearly DC values for both ${\hat{V}}_{\mathrm{Cdq}}$ and ${\hat{I}}_{\mathrm{dq}}$ at steady state. Under this effect, the observer system plant G_sys_(s) in Figure 6 is obtained by
(21)$$\left[\begin{array}{c} G_{sys\_d}(s)\\ G_{sys\_q}(s) \end{array}\right]=\left[\begin{array}{c} \frac{{\hat{V}}_{Cd2}}{V_{CFF\_Id}^{*}+V_{dis\_d}}\\ \frac{{\hat{V}}_{Cq2}}{V_{CFF\_Iq}^{*}+V_{dis\_q}} \end{array}\right]=\frac{1}{{\hat{L}}_{f}{\hat{C}}_{p}s^{2}+{\hat{R}}_{f}{\hat{C}}_{p}s+(1-{\hat{\omega }}_{e}^{2}{\hat{L}}_{f}{\hat{C}}_{p})}\left[\begin{array}{c} 1\\ 1 \end{array}\right]$$

Considering the design of the observer controller C_Io_(s) in Figure 6, the purpose is to achieve no steady state error for the closed-loop voltage regulation. Thus, this is given by
(22)$$C_{Io}(s)=\left[\begin{array}{c} {\hat{V}}_{dis\_d}/\left[{\hat{V}}_{Cd}-{\hat{V}}_{Cd2}\right]\\ {\hat{V}}_{dis\_q}/\left[{\hat{V}}_{Cq}-{\hat{V}}_{Cq2}\right] \end{array}\right]=K_{p\_I}\frac{{\hat{L}}_{f}{\hat{C}}_{p}s^{2}+{\hat{R}}_{f}{\hat{C}}_{p}s+(1-{\hat{\omega }}_{e}^{2}{\hat{L}}_{f}{\hat{C}}_{p})}{s}\left[\begin{array}{c} 1\\ 1 \end{array}\right]$$
where K_p_I_ is the proportional gain used to determine the overall observer estimation bandwidth. By substituting G_sys_(s) in Equation (21) and C_Io_(s) in Equation (22) into Figure 6, this current observer can be simplified as shown in Figure 7.

As seen in Figure 7, the overall observer dynamic property is equivalent to a first-order low-pass filter where the bandwidth is dependent on the proportional gain K_p_I_. Considering the estimation errors from the PWM inverter, K_p_I_ is set at 100 Hz to remove these high-frequency noises.

Similar to the capacitor voltage observer mentioned in part A, ${\hat{V}}_{\mathrm{dis}\_\mathrm{dq}}$ in Equation (19) represents the internal disturbances for the proposed current observer. Considering the ideal observer regulation without steady state error, ${\hat{V}}_{\mathrm{dis}\_\mathrm{dq}}$ should appear at the signal node illustrated in Figure 6. It is important that ${\hat{V}}_{\mathrm{dis}\_\mathrm{dq}}$ contains the desired estimator of machine current ${\hat{I}}_{\mathrm{dq}}$. In this case, ${\hat{I}}_{\mathrm{dq}}$ can be eventually obtained from Equation (23). This is shown by
(23)$$\left[\begin{array}{c} {\hat{I}}_{d}\\ {\hat{I}}_{q} \end{array}\right]=\frac{-1}{{\hat{R}}_{f}^{2}+{\left({\hat{L}}_{f}{\hat{\omega }}_{e}\right)}^{2}}\left[\begin{array}{cc} {\hat{R}}_{f} & {\hat{L}}_{f}{\hat{\omega }}_{e}\\ -{\hat{L}}_{f}{\hat{\omega }}_{e} & {\hat{R}}_{f} \end{array}\right]\left[\begin{array}{c} {\hat{V}}_{dis\_d}\\ {\hat{V}}_{dis\_q} \end{array}\right]=\left[\begin{array}{c} G_{cal\_d}\\ G_{cal\_q} \end{array}\right]\left[\begin{array}{c} {\hat{V}}_{dis\_d}\\ {\hat{V}}_{dis\_q} \end{array}\right]$$
where C_cal_ in Figure 6 is the ratio between ${\hat{V}}_{\mathrm{dis}\_\mathrm{dq}}$ and ${\hat{I}}_{\mathrm{dq}}$. Besides, the capacitor current ${\hat{I}}_{\mathrm{Cdq}}$ can be simply calculated by ${\hat{I}}_{\mathrm{Cdq}}\;=\;I_{\mathrm{Ldq}}\;-\;{\hat{I}}_{\mathrm{dq}}$. In contrast to the direct capacitor current estimation in Equation (14), it is interesting to evaluate the estimation accuracy for the proposed closed-loop current observer. Based on Equation (17), ${\hat{I}}_{\mathrm{dq}}$ and ${\hat{I}}_{\mathrm{Cdq}}$ are both estimated using the estimated capacitor voltage ${\hat{V}}_{\mathrm{Cdq}}$ and LC filter parameters. To easily compare the current estimation error based on the direct estimation in Equation (15), only the capacitor current estimation error I_Cdq_err_ is calculated based on the proposed voltage observer in Equation (10) and current observer model in Equation (17). Through the mathematical organization, the estimation accuracy of the proposed current observer in Figure 6 is given by
(24)$$\begin{array}{cc} \left[\begin{array}{l} I_{Cd\_err}\\ I_{Cq\_err} \end{array}\right]= & \frac{K_{p\_I}}{s+K_{p\_I}}\frac{-1}{{\hat{R}}_{f}^{2}+{\left({\hat{L}}_{f}{\hat{\omega }}_{e}\right)}^{2}}\{C\left[\begin{array}{l} V_{d}^{*}\\ V_{q}^{*} \end{array}\right]+D\left[\begin{array}{c} V_{q}^{*}\\ -V_{d}^{*} \end{array}\right]\\ & +\left(-C\left(R_{f}+L_{f}s\right)+DL_{f}\omega_{e}\right)\left[\begin{array}{c} I_{Ld}\\ I_{Lq} \end{array}\right]+\left(-D\left(R_{f}+L_{f}s\right)-CL_{f}\omega_{e}\right)\left[\begin{array}{c} I_{Lq}\\ -I_{Ld} \end{array}\right]\} \end{array}$$
For simplification, two variables **C** and **D** in Equation (24) are defined by
(25)$$\left[\begin{array}{c} C\\ D \end{array}\right]=\left[\begin{array}{c} {\hat{R}}_{f}\left(A-\hat{A}\right)-{\hat{L}}_{f}{\hat{\omega }}_{e}\left(B-\hat{B}\right)\\ {\hat{R}}_{f}\left(B-\hat{B}\right)+{\hat{L}}_{f}{\hat{\omega }}_{e}\left(A-\hat{A}\right) \end{array}\right]$$
where **A** and **B** are
(26)$$\left[\begin{array}{c} A\\ B \end{array}\right]=\left[\begin{array}{c} L_{f}C_{p}s^{2}+R_{f}C_{p}s+\left(1-\omega_{e}^{2}L_{f}C_{p}\right)\\ R_{f}C_{p}\omega_{e} \end{array}\right]$$
In Equation (25), $\hat{A}$ and $\hat{B}$ are defined in the same manner as **A** and **B**; however, the estimated parameters ${\hat{L}}_{f}$, ${\hat{R}}_{f}$, ${\hat{C}}_{p}$ and ${\hat{\omega }}_{e}$ are substituted. Similar to the open-loop estimation error of capacitor current I_Cdq_err_ in Equation (15), the current estimation error based on the proposed observer is investigated with respect to I_Ldq_ and $V_{\mathrm{dq}}^{*}$. Figure 5 illustrates the estimation accuracy of the proposed current observer under the parameter variation of ${\hat{L}}_{f}$ and ${\hat{C}}_{p}$. In Figure 5a,b, |I_Cdq_err_(jω)/I_Ld_(jω)| and |I_Cdq_err_(jω)/I_Lq_(jω)| with respect to the inductance ${\hat{L}}_{f}$ are analyzed. Compared to the direct current estimation, the proposed current observer is able to reduce the sensitivity error of ${\hat{L}}_{f}$ in the whole frequency range. Besides, in Figure 5c, the variation of |I_Cdq_err_(jω)/$V_{d}^{*}$(jω)| with respect to ${\hat{C}}_{p}$ is investigated. In contrast to direct current estimation, better estimation performance with respect to ${\hat{C}}_{p}$ error is achieved due to the closed-loop observer estimation.

## 5. Experiment Results

This section experimentally verifies the proposed dual-observer estimation on a low-inductance PM machine drive with an LC filter. Figure 8 shows the test setup of the PM machine with an LC filter. A 105 W PM machine with only 130 μH phase inductance was tested. Detailed motor and filter specifications are listed in Table 1. Considering the high-speed limitation, different fans were attached on the motor shaft instead of servo machines for the torque load operation.

### 5.1. LC Filter on Current Harmonics

This part verifies the improvements of the PWM reflected current harmonics using the LC filter. Figure 9 compares the machine phase currents and the corresponding spectrum for the drive (a) without and (b) with the LC filter. The machine speed was operated at 40 krpm (666.7 Hz electric frequency) through the FOC with external current sensors. The phase currents I_A_/I_B_/I_C_ in Figure 1 were measured using current probes. For this low-inductance machine drive without an LC filter in (a), the PWM switch caused considerable current harmonics with 65.1% total harmonic distortion (THD). PWM-reflected rotor loss and a high rotor temperature were also observed. In contrast, in (b) with the LC filter, these current harmonics were reduced with only 20% THD. Based on this test, it is concluded that the LC filter effectively improves the PWM-reflected current harmonics on the FOC drive.

### 5.2. LC Filter Effect on FOC Drive

This part demonstrates the influence of the LC filter on the FOC machine drive. Figure 10 compares three different A-phase currents—the inverter output current, I_LA_, capacitor current I_CA_ and machine actual current I_A_ in Figure 1, respectively—at (a) a low speed of 1 krpm and (b) high speed of 40 krpm. As seen in Figure 10a, the inverter current I_LA_ and machine current I_A_ were nearly the same, while the capacitor current I_CA_ contained some PWM harmonics. Besides, current waveform distortion was observed on both I_LA_ and I_A_. In general, the PWM inverter contained the deadtime effect, affecting the current regulation, especially at low speed. In this paper, the dead-time compensation was applied based on the direct voltage compensation dependent on the phase current polarity [25]. In contrast, at high speed, as shown in Figure 10b, a visible magnitude and phase difference between I_LA_ and I_A_ were observed due to the high magnitude of I_CA_. Based on this experiment, it is concluded that, at high speed, the actual machine current I_A_ must be obtained to stabilize the FOC drive.

### 5.3. Capacitor Voltage Estimation

This section evaluates the capacitor voltage estimation using the proposed capacitor voltage observer in Figure 4. Figure 11 shows the comparison of the actual dq capacitor voltages V_Cdq_ and estimated capacitor voltages ${\hat{V}}_{\mathrm{Cdq}}$. Here, V_Cdq_ values were measured using external voltage sensors. In this test, the motor was accelerated from 6 krpm to 42 krpm within 4 s. The FOC drive was applied with external current sensors to clearly evaluate the estimation performance. In this test, the sensorless FOC was implemented where the rotor position was obtained from the EMF estimation. Because the EMF is insufficient at very low speed, the proposed FOC drive with an LC filter was implemented at 6 krpm with accurate EMF estimation. As seen in Figure 11, ${\hat{V}}_{\mathrm{Cdq}}$ was close to V_Cdq_ n the range of 6~42 krpm under different voltage magnitudes. It is concluded that the proposed voltage observer can obtain actual capacitor voltages at different speeds and magnitudes using only the existing drive current sensors.

### 5.4. Motor Current Estimation

This section shows the machine current estimation using the proposed capacitor current observer in Figure 6. It is noted that both the machine current I_dq_ and capacitor current I_Cdq_ can be obtained from this observer. In this context, only the motor ${\hat{I}}_{\mathrm{dq}}$ estimation performance is evaluated for simplicity.

Figure 12 compares the actual machine I_dq_ and estimated ${\hat{I}}_{\mathrm{dq}}$ using the proposed current observer. The test condition was the same as Figure 11 in terms of the acceleration with the FOC drive. As shown in Figure 8, an external fan was coupled to the machine shaft to easily manipulate the torque load. This machine drive test setup was similar to [26]. In general, the torque load is a quadratic function dependent on the machine speed. At 6 krpm, as shown in Figure 12, relatively high current noises were observed. However, the current noises were negligible as the speed increased since the magnitude of noise remained the same irrespective of the speed.

More importantly, ${\hat{I}}_{\mathrm{dq}}$ was close to I_dq_ irrespective of the machine speed and current magnitude. As seen in Figure 12, q-axis current estimation error was around 2.56% at 6 krpm. This increased to 4.54% as the speed increased to 42 krpm. In contrast, the d-axis current estimation error was negligible for the speeds between 6 krpm and 42 krpm. As a result, the proposed current observer was also able to estimate ${\hat{I}}_{\mathrm{dq}}$ accurately at different speeds once the current magnitude was higher than a certain value.

Figure 13 further illustrates the time-domain waveforms of machine AB-line voltage ${\hat{V}}_{\mathrm{AB}}$ and machine A-phase current ${\hat{I}}_{A}$ at a low speed of 1 krpm (16.67 Hz) with respect to the externally measured V_AB_ and I_A_. Both (a) a 25% load and (b) a 50% load were compared based on different fan load sizes. For the voltage estimation, ${\hat{V}}_{\mathrm{AB}}$ was close to V_AB_ at different loads. It is noted that the filter cutoff frequency was designed at 2.92 kHz. In this case, the PWM voltage could be removed in both V_AB_ and ${\hat{V}}_{\mathrm{AB}}$, leading to nearly sinusoidal voltage waveforms. In contrast, for current estimation, a relatively low signal-to-noise ratio was observed in ${\hat{I}}_{A}$. As mentioned in Equation (18), the machine current was estimated from voltage disturbances ${\hat{V}}_{\mathrm{dis}\_\mathrm{dq}}$ through the RLC circuit model. This voltage was relatively small at low speed, resulting in estimation noises. However, the magnitudes between ${\hat{I}}_{A}$ and I_A_ were almost the same. It is concluded that the proposed machine current estimation can still be used for FOC current regulation.

### 5.5. Sensitivity Analysis of Filter State Estimation

This section further verifies the influence of filter parameter errors on the proposed observer-based estimation. As mentioned in Figure 5, the estimation of the filter output machine current ${\hat{I}}_{\mathrm{dq}}$ requires a filter capacitor and inductance parameter. More importantly, these parameter errors can be reduced using the proposed observer with the closed-loop feedback regulation.

Figure 14 shows the time-domain waveforms of the measured speed, measured and estimated machine I_q_ and ${\hat{I}}_{q}$ and the estimation error $I_{q}\;-\;{\hat{I}}_{q}$. The test condition was the same as in Figure 11 and Figure 12 in terms of the FOC acceleration. In (a), a 50% capacitor parameter error was intentionally added in the proposed current observer. The current estimation error was negligible as the speed accelerated from 6 krpm to 42 krpm. Thus, the proposed current observer demonstrated negligible sensitivity under the filter capacitor variation. This test result is consistent with the analytical model in Figure 5c.

In contrast, in Figure 14b, the same 50% filter inductance error was included to evaluate the same ${\hat{I}}_{q}$ estimation performance. In contrast to the capacitor parameter, the estimation error increased as the speed increased. At 6 krpm, the current estimation error was negligible. However, this error increased to 26.3% as the speed increased to 42 krpm. This result was the same as the simulation in Figure 5a. Nevertheless, the filter inductance parameter should be known as it is selected by the drive designer.

### 5.6. FOC Performance Comparison

The machine speed closed-loop control for the drive with an LC filter is compared in this section. Figure 15 compares the current regulation using the feedback of (a) the inverter current I_Ldq_ in Figure 1, (b) the estimated machine current ${\hat{I}}_{\mathrm{dq}}$ with direct estimation in Equation (14) and (c) the estimated machine current ${\hat{I}}_{\mathrm{dq}}$ with proposed observer-based estimation in Figure 6. Due to the encoder installation issue on the test machine, the EMF-based position estimation was applied where the estimated capacitor voltage ${\hat{V}}_{\mathrm{Cdq}}$ was used for the position estimation, as illustrated in Figure 2.

As shown in Figure 15a, if the inverter current I_Ldq_ was directly used for current regulation ignoring the resonant dynamics of the LC filter, the speed control ultimately failed when the speed reached 40 krpm (666.7 Hz frequency). The difference between I_Ldq_ and I_dq_ at high speed was the primary issue. In contrast, in (b) when the directly estimated ${\hat{I}}_{\mathrm{dq}}$ was used, the speed control was able to stably operate at the rated speed under fan load. However, current noises were observed. These current noises resulted in speed estimation noises with 5.5% error in the high-speed region. More importantly, in Figure 15c, better current regulation performance compared to (b) was achieved using the capacitor current observer for ${\hat{I}}_{\mathrm{dq}}$ estimation. In particular, at a high-speed of 42 krpm, there was no evident noise in the speed estimation error. For both direct estimation and proposed dual-observer, there was a 5.5% error in speed estimation at 6 krpm due to the low noise-to-signal ratio of back-EMF voltage. Thus, the proposed dual-observer could accurately obtain ${\hat{V}}_{\mathrm{Cdq}}$ and ${\hat{I}}_{\mathrm{dq}}$ for the position sensorless FOC drives under the PWM drive with an LC filter.

## 6. Conclusions

This paper proposes a dual-observer to estimate the capacitor voltage/current and machine current for a sensorless FOC drive using the drive inside current sensors. The observer estimation performance considering the filter parameter sensitivity is fully investigated. Table 2 lists the performance comparison of the conventional model-based estimation and proposed dual-observer estimation. Compared to conventional model-based estimation, the proposed dual-observer demonstrates robust estimation performance under parameter errors. The capacitor parameter error results in a negligible influence on the proposed observer estimation. The filter inductance error only affects the capacitor current estimation at high speed. By using commercial PWM drives, the FOC can be applied to PM machines where an LC filter is added to reduce the PWM harmonics.

## Figures and Tables

**Figure 1 sensors-21-03596-f001:**
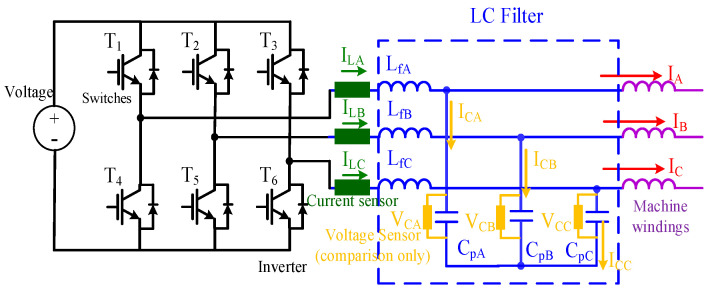
Machine drive with output LC filter [21].

**Figure 2 sensors-21-03596-f002:**
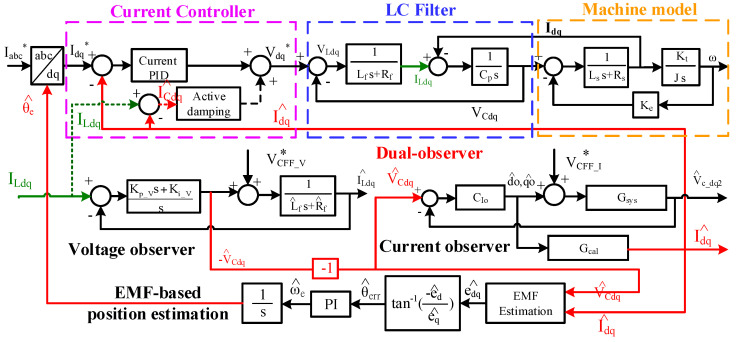
Illustration of FOC machine drive with LC filter using proposed dual-observer for the filter voltage and current estimation.

**Figure 3 sensors-21-03596-f003:**
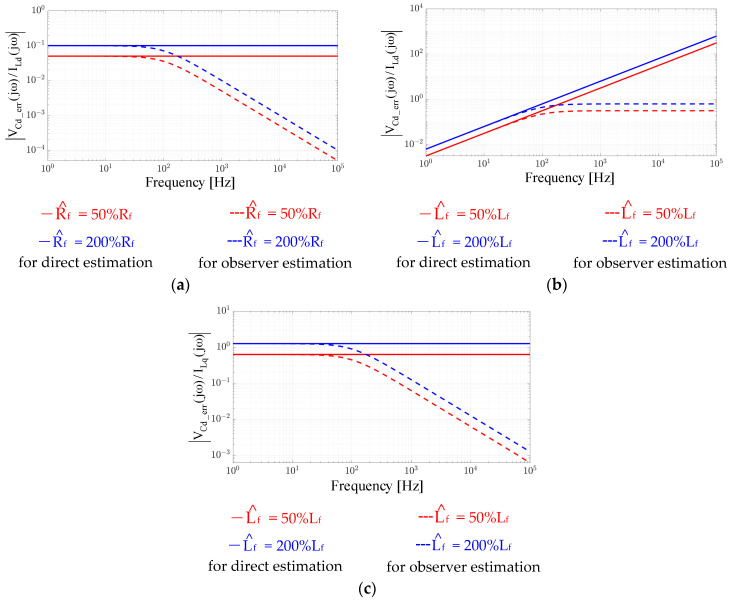
Estimation accuracy evaluation of ${\hat{V}}_{\mathrm{Cdq}}$ between direct estimation in (4) and proposed observer in Equation (9): (**a**) ${\hat{R}}_{f}$ error on |V_Cd_err_(jω)/I_Ld_(jω)|, (**b**) $\hat{L_{f}}$ error on |V_Cd_err_(jω)/I_Ld_(jω)|, and (**c**) ${\hat{L}}_{f}$ error on |V_Cd_err_(jω)/I_Lp_(jω)|.

**Figure 4 sensors-21-03596-f004:**
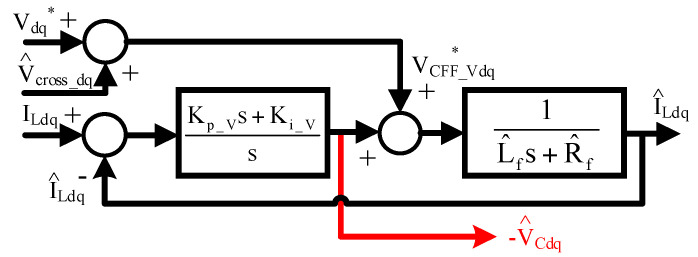
Proposed first closed-loop observer in Figure 2 for capacitor voltage estimation (capacitor voltage observer).

**Figure 5 sensors-21-03596-f005:**
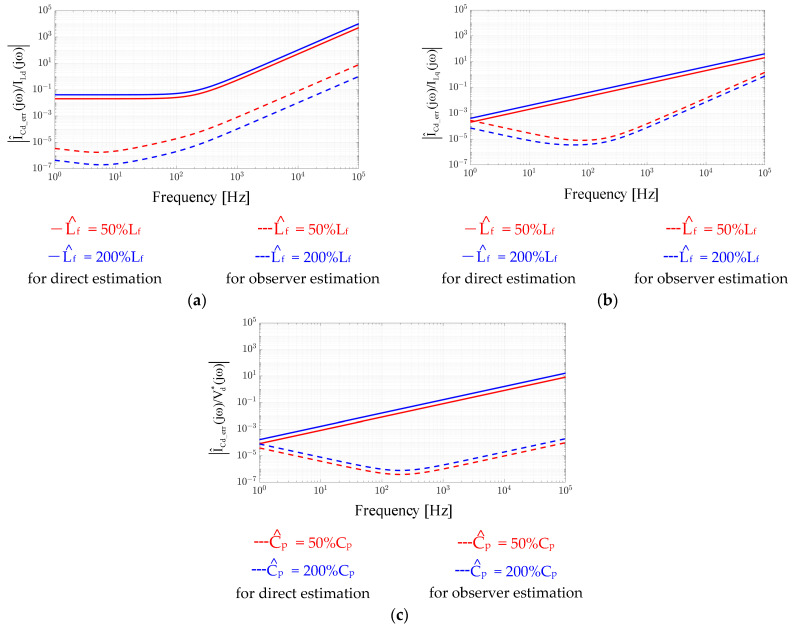
Estimation accuracy evaluation of ${\hat{I}}_{\mathrm{Cdq}}$ between direct estimation in Equation (13) and proposed observer in Figure 6: (**a**) ${\hat{L}}_{f}$ error on |I_Cd_err_(jω)/I_Ld_(jω)|, (**b**) ${\hat{L}}_{f}$ error on |I_Cd_err_(jω)/I_Lq_(jω)|, and (**c**) ${\hat{C}}_{p}$ error on |I_Cd_err_(jω)/$V_{d}^{*}$(jω)|.

**Figure 6 sensors-21-03596-f006:**
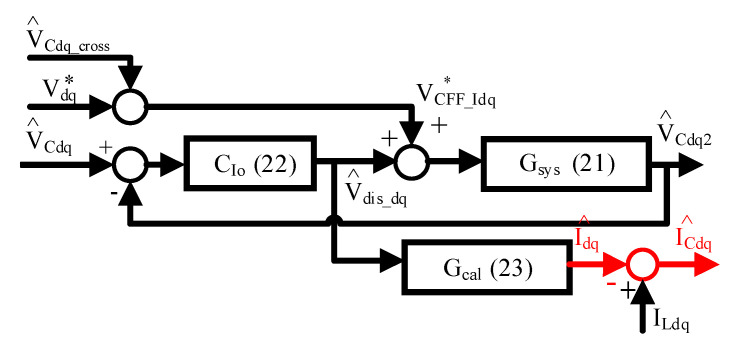
Proposed second observer in Figure 2 for capacitor current estimation and machine current estimation.

**Figure 7 sensors-21-03596-f007:**
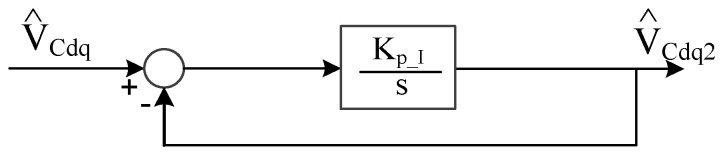
Simplification of capacitor current observer in Figure 6.

**Figure 8 sensors-21-03596-f008:**
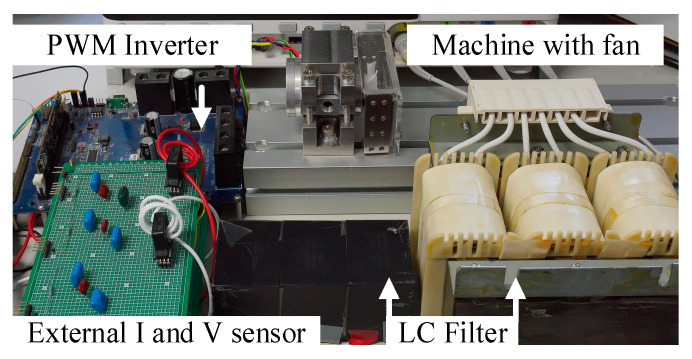
Test setup of PM motor drive with LC filter.

**Figure 9 sensors-21-03596-f009:**
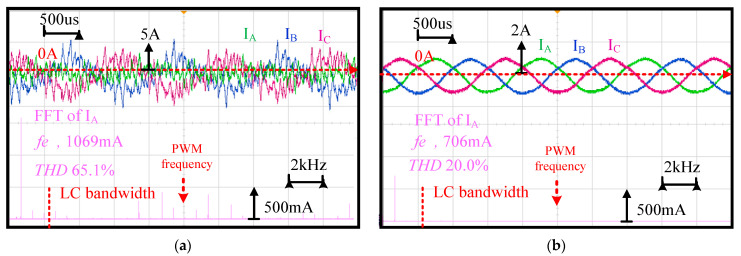
Machine three-phase currents and the corresponding spectrum for the drive (**a**) without and (**b**) with LC filter (40 krpm and fan load).

**Figure 10 sensors-21-03596-f010:**
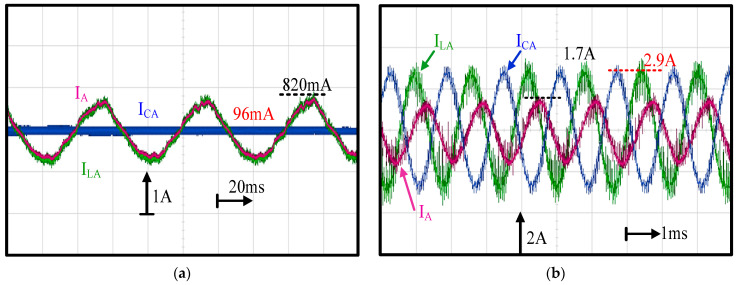
Experimental comparison of A-phase inverter output current I_LA_ capacitor current I_CA_ and machine actual current I_A_ at (**a**) a low speed of 1 krpm and (**b**) a high speed of 40 krpm (FOC drive and fan load).

**Figure 11 sensors-21-03596-f011:**
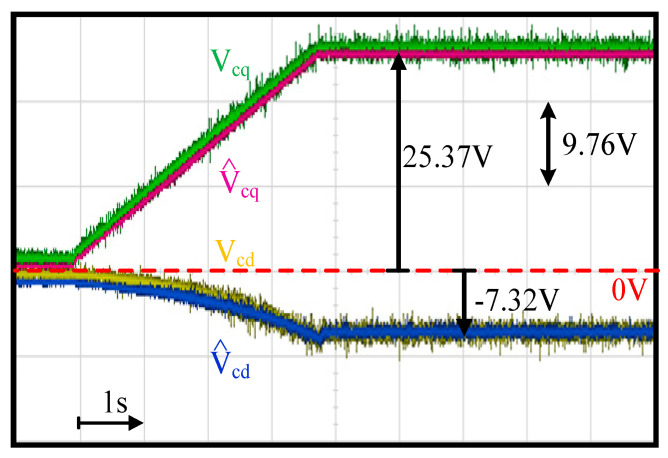
Experiment of capacitor voltage estimation performance (acceleration from 6 krpm to 42 krpm within 4 s and fan load).

**Figure 12 sensors-21-03596-f012:**
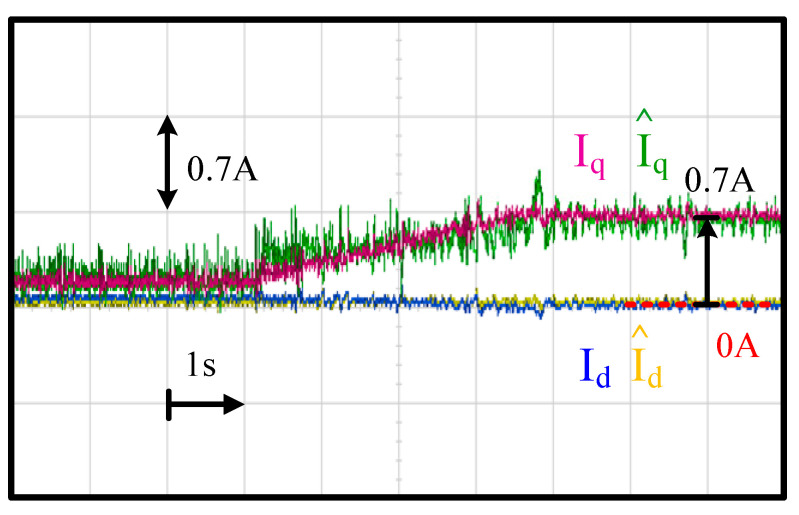
Experiment of machine current estimation performance (acceleration from 6 krpm to 42 krpm within 4 s and fan load).

**Figure 13 sensors-21-03596-f013:**
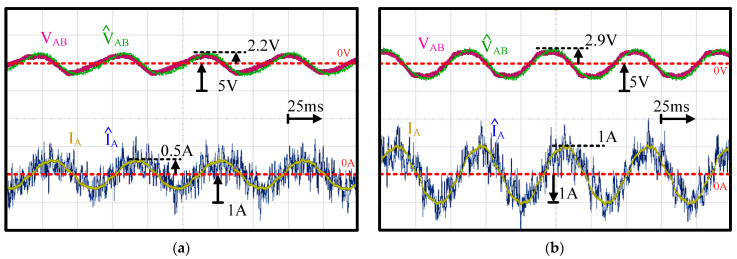
Experiment of estimated filter output capacitor voltage and machine current at (**a**) 25% and (**b**) 50% load (1 krpm rotor speed).

**Figure 14 sensors-21-03596-f014:**
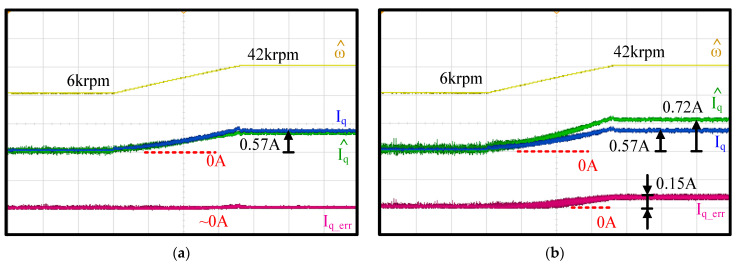
Sensitivity analysis of proposed observer-based current estimation under (**a**) filter capacitor error and (**b**) inductance error (6 krpm to 42 krpm and fan load).

**Figure 15 sensors-21-03596-f015:**
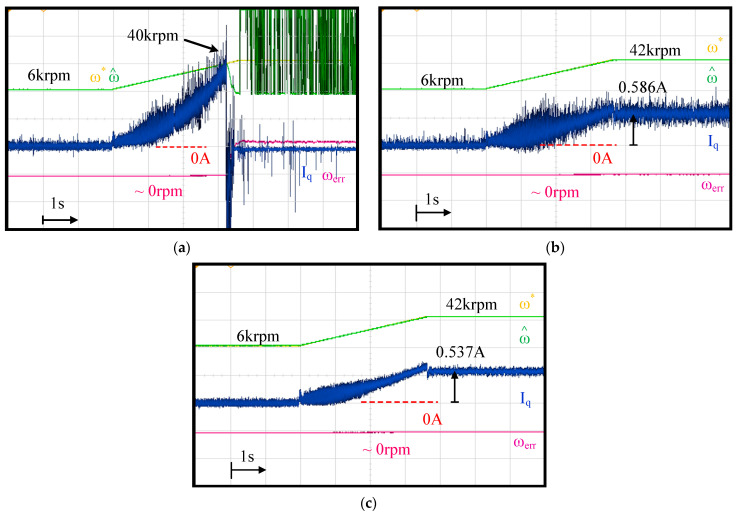
Comparison of current regulation using (**a**) inverter current I_Ldq_, (**b**) estimated machine current ${\hat{I}}_{\mathrm{dq}}$ with direct estimation in (14) and (**c**) estimated machine current ${\hat{I}}_{\mathrm{dq}}$ with proposed observer-based estimation in Figure 6.

**Table 1 sensors-21-03596-t001:** Test PM machine characteristics.

**Test Machine**	**Values**
Rotor poles	2-pole
Rated torque	25 mNm
Rated current	2 A
Rated speed	42 krpm (700 Hz elec. frequency)
Phase resistance	0.85 Ω
Phase inductance	130 μH
DC bus voltage	40 V
**LC Filter**	**Values**
Inductance	1 mH
Capacitor	25.8 μF
Resonant frequency	2.92 kHz

**Table 2 sensors-21-03596-t002:** Comparison between conventional model-based estimation and proposed dual-observer estimation.

	Model-Based Estimation	Proposed Dual-Observer
Parameter sensitivity	Capacitor and inductor sensitive (drive is unstable for experimental test)	Capacitor (negligible error) Inductor (negligible error at 6 krpm and 26.3% error at 42 krpm)
FOC speed control	5.5% error between 6~42 krpm	5.5% error at 6 krpm and negligible error at 42 krpm
Memory usage	14.2 Kb	18.5 Kb

## Data Availability

Data sharing is not applicable to this article.

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
