# Peer review of "Sensorless LC Filter Implementation for Permanent Magnet Machine Drive Using Observer-Based Voltage and Current Estimation"

_sensors, 2021, doi:10.3390/s21113596_

Round 1
Reviewer 1 Report
The work is well done, reads fluently and clearly describes the method used, starting from the abstract.
The introduction is very clear and explains well what problem is solved and with what innovation, even from the point of view of contextualization is well done and properly accompanied by extensive bibliography. 
Paragraph 2 is very interesting with the due clarifications on the sizing of the system, the work is well introduced and explained. It allows the reader to better enter in the theme.
The calculation method is clearly described. Everything is clearly validated and further explained through the description of the tests in paragraph 5.
Only the conclusions are slightly lacking, I expected a brief discussion with general data results of theoretical and experimental tests. It would be better to explain with a summary chart what the improvement obtained is worth. It would be also interesting a table with the merits and defects of the proposed system, this to give more evidence of the quality of the improvement presented here. 
Author Response
Please review the attached word file for more detail.

Reviewer 2 Report
In this manuscript, the authors presented a dual observer topology to estimate filter voltage and current for sensorless field-orientated control drives. Some suggestions are as follows:
- In the abstract, authors are advised to clearly highlight their contributions in this paper, briefly describe their procedures, and provide more quantitative comparison results.
- For Figure 1 that has been published before, authors are advised to cite the reference in the caption.
- In the experimental sections 5.4 to 5.6, authors present figures and results without precise and quantitative evaluations. Authors shall give a more critical analysis.
- The overall conclusion also needs to be rewritten, and further supported by quantitative comparisons with other methods.
- Some typos were observed in the manuscript:
- “Figures 12(c)” in line 390
- “researches” in line 386

Author Response

(The authors gave the same response as above.)

Round 2
Reviewer 2 Report
The provided suggestions have been properly addressed by the authors.